# Thermophilic Inorganic Pyrophosphatase Ton1914 from *Thermococcus onnurineus* NA1 Removes the Inhibitory Effect of Pyrophosphate

**DOI:** 10.3390/ijms232112735

**Published:** 2022-10-22

**Authors:** Yajing Li, Xue Yang, Renjun Gao

**Affiliations:** Key Laboratory for Molecular Enzymology and Engineering of Ministry of Education, School of Life Science, Jilin University, Changchun 130021, China

**Keywords:** pyrophosphatases, *Thermococcus onnurineus*, UDP-glucose, UDP-galactose, polymerase chain reaction

## Abstract

Pyrophosphate (PP_i_) is a byproduct of over 120 biosynthetic reactions, and an overabundance of PP_i_ can inhibit industrial synthesis. Pyrophosphatases (PPases) can effectively hydrolyze pyrophosphate to remove the inhibitory effect of pyrophosphate. In the present work, a thermophilic alkaline inorganic pyrophosphatase from *Thermococcus onnurineus* NA1 was studied. The optimum pH and temperature of Ton1914 were 9.0 and 80 °C, respectively, and the half-life was 52 h at 70 °C and 2.5 h at 90 °C. Ton1914 showed excellent thermal stability, and its relative enzyme activity, when incubated in Tris-HCl 9.0 containing 1.6 mM Mg^2+^ at 90 °C for 5 h, was still 100%, which was much higher than the control, whose relative activity was only 37%. Real-time quantitative PCR (qPCR) results showed that the promotion of Ton1914 on long-chain DNA was more efficient than that on short-chain DNA when the same concentration of templates was supplemented. The yield of long-chain products was increased by 32–41%, while that of short-chain DNA was only improved by 9.5–15%. Ton1914 also increased the yields of UDP-glucose and UDP-galactose enzymatic synthesis from 40.1% to 84.8% and 20.9% to 35.4%, respectively. These findings suggested that Ton1914 has considerable potential for industrial applications.

## 1. Introduction

Pyrophosphate (PP_i_) is a byproduct of a variety of reversible nucleoside 5′-triphosphate-dependent processes. Excessive PP_i_ damages organism growth and development and lowers the yield of nucleoside transfer events [1]. After being degraded to phosphate (P_i_) by pyrophosphatase (PPases, EC3.6.1.1), hydrolyzed pyrophosphate re-enters metabolic pathways that relate to various energy-demanding biochemical transformations, such as DNA replication, protein synthesis, and lipid metabolism [2,3]. Soluble PPases are grouped into three structurally distinct families: Family I, Family II, and Family III [4]. Family I PPases are widely distributed in all living cells, whereas Family II and Family III PPases are found only in prokaryotes [5]. PPases are metal-dependent hydrolases, and alkaline PPase relies on the assistance of Mg^2+^ to present a high affinity for pyrophosphate and turn on catalysis [6]. PPases have become the most promising candidates for industrial and pharmaceutical applications in recent years. For example, PPases are used to boost the yield of pyrophosphate hydrolysis reactions, such as nucleotide sugar synthesis in industrial applications [7,8]. As PCR enhancers, PPases improve the efficiency of the polymerase chain reaction [9] and contribute to the accuracy of gene sequencing.

Because of its high sensitivity, good specificity, and low purity requirements, polymerase chain reaction (PCR) is commonly used to amplify DNA in vitro, and it plays an important role in biological research, clinical treatments, and criminal investigations [10]. Nevertheless, the limitations of PCR, such as low fidelity and byproduct inhibition, have a great impact on PCR productivity; thus, the use of PCR enhancers has become a major focus of research. There are three typical options: 1. Degrading the byproducts to remove the inhibition and increase the reaction rate (dITPase, dUTPase, PPases) [9,11,12]; 2. Accelerating the effective collision between the reactants by increasing the temperature quickly (graphene nanoflakes) [13]; and 3. Safeguarding and maintaining the DNA polymerases and template DNA structures (detergents, DMSO) [14,15,16]. PPases have been shown to be helpful for driving the equilibrium of the DNA elongation reaction because they hydrolyze PP_i_ to relieve the inhibition of PCR. Even some DNA polymerases containing an additional polymerase-histidinol-phosphatase (PHP) domain confirm that they play a functional role in exerting pyrophosphatase activity [17,18]. As a result, DNA replication can be carried out more effectively when DNA polymerases and pyrophosphatases are combined. However, the high temperature of PCR reduces or even inactivates the activity of PPases, and it is crucial to exploit neotypic heat-resistant PPases.

In glycobiology, sugar nucleotides play a significant role as an intermediary. UDP-glucose and UDP-galactose are the most extensively used substrates for the synthesis of glycosides, oligosaccharides, and polysaccharides, and they are essential in physiological metabolism. However, the low stability of UDP-sugar makes it difficult and time-consuming to synthesize in industrial production, which results in expensive and scarce products. This is an important issue that needs to be solved. There are two main methods of synthesizing sugar nucleotides: chemical methods and enzymatic methods. NTP-sugar can be produced through chemical synthesis; however, scaled-up manufacturing procedures are not economically feasible because of the low stereoselectivity, laborious process, and expensive cost [19]. Enzymatic approaches may circumvent the drawbacks of chemical synthesis and are more in line with the mainstream market due to their high stereoselectivity, environmental compatibility, and energy-saving properties [20]. Moreover, PPases can alleviate PP_i_ inhibition during the enzymatic synthesis of nucleotide sugars, thereby increasing yields and making these enzymes more suitable for industrial production. 

In this study, we cloned and expressed the thermophilic alkaline inorganic pyrophosphate Ton1914 from *Thermococcus onnurineus* NA1. The results suggested that Ton1914 had excellent thermal stability and showed great application prospects in increasing sugar nucleotide yield and accelerating the PCR process.

## 2. Results

### 2.1. Molecular Properties of Ton1914

Soluble pyrophosphatases are generally oligomeric proteins consisting of several identical subunits. Eukaryotic pyrophosphatases are homomeric structures composed of approximately 30 kDa subunits [21]. Bacterial and archaeal pyrophosphatases are hexametric or tetrameric complexes formed by subunits of approximately 20 kDa [22]. The Ton1914 gene is 537 bp and encodes 178 amino acids. The molecular weight of Ton1914 is approximately 25 kDa, as estimated by SDS–PAGE (Figure 1). This finding is not consistent with the size calculated from the amino acid sequence (20.8 kDa). *Ph*PPases [23] also showed this phenomenon, and Ton1914 was conjectured to carry positive charges and only bind a small amount of SDS in SDS–PAGE, resulting in slower protein movement and higher molecular weight than the theoretical. Native-PAGE results illustrated that the molecular mass of Ton1914 is approximately 120 kDa. Additionally, we modeled its structure using the Discovery Studio, and the results showed that it is a homo-hexamer (Appendix A).

### 2.2. Catalytic Properties of Ton1914

The optimum temperature and pH of Ton1914 were assayed in 50 mM Tris-HCl buffer. The results demonstrated that the enzymatic activity of Ton1914 was less than 1% of the maximum activity below 50 °C. The enzyme activity increased significantly as the temperature increased and reached a maximum of 80 °C (Figure 2a). Ton1914 is an alkaline protein with a narrow pH range that shows high activity in the pH range of 8.0 to 9.5, peaking at pH 9.0. The enzyme activity decreased sharply outside this range (Figure 2b). Numerous studies have shown that both Family I and Family II PPases show strong metal cation dependence, and Family I PPases exhibit the highest dependence on Mg^2+^ and sensitivity to Ca^2+^ inhibition [24]. Nevertheless, *Lms*PPase from the trypanosomatid *Leishmania major* was an exception and preferred Ca^2+^ over Mg^2+^ [25]. In contrast, Family II PPases prefer Mn^2+^ (or Co^2+^) to Mg^2+^ [24]. PPases binding three Mg^2+^ ions and one transition metal ion (Mn^2+^, Co^2+^) are the key elements of catalytic activation [5]. Ton1914 belongs to Family I PPases, and only Mg^2+^ could distinctly activate the enzyme activity. The activity in the presence of Mn^2+^ was only 20% of that with the addition of Mg^2+^, and other metal cations did not work as cofactors (Figure 2c). Furthermore, Ton1914 responded sensitively to the concentration of Mg^2+^ and was almost inactivated in the absence of Mg^2+^, and the activity increased with increasing Mg^2+^, with the greatest activity of 389.3 U/mg reached at 1.6 mM. The activity gradually decreased when the concentration of Mg^2+^ was over 1.6 mM (Figure 2d). The inhibition at high cation concentrations was probably due to the presence of excessive free cations that destroyed the three-dimensional structure of proteins and diminished their enzymatic activity [23]. The above results were approximately identical to published findings for alkaline thermophilic inorganic pyrophosphatases.

### 2.3. pH, Thermal and Metal Stability of Ton1914

Ton1914 had excellent thermal stability and retained 82.3% of its activity after incubation at 70 °C for 20 h. The half-lives of the recombinant enzyme were 7 h and 2.5 h at 80 °C and 90 °C, respectively (Figure 3a). Yan [6] and Jeon [23] demonstrated that the addition of Mg^2+^ enhances the thermal stability of PPases, which is attributed to the fact that a suitable amount of Mg^2+^ can stabilize the structure of the enzyme and make it less susceptible to denaturation in a high-temperature environment. Although the thermal stability of Ton1914 was not the best among the reported thermophilic PPases (Table 1), it was also perfectly suitable for high-temperature reactions and maintained high activity. In a test of the effect of Mg^2+^ on enzyme stability, we found that the coincubation of Mg^2+^ and Ton1914 could significantly improve the thermal stability of Ton1914. With the extension of coincubation time, the relative enzyme activity initially increased and then decreased, and the highest enzyme activity reached 119% after incubation for 3 h with 1.6 mM Mg^2+^. Surprisingly, the enzyme activity of Ton1914 after incubation with 1.6 mM Mg^2+^ at 90 °C for 5 h was still 100% (Figure 3b). We asserted that moderate amounts of Mg^2+^ could increase enzyme activity through binding to the enzyme to enhance the structural rigidity of the enzyme and refolding to the most stable enzyme structure takes a certain amount of time. In terms of pH stability, Ton1914 showed good stability at pH 7.0–10.0, with approximately 80% relative activity after 24 h of enzyme incubation (Figure 3c).

### 2.4. Substrate Specificity of Ton1914

Inorganic pyrophosphatases have extremely high hydrolytic activity toward their natural substrate pyrophosphate, and numerous studies have shown that PPases also catalyze a small amount of hydrolysis on tripolyphosphate (PPP_i_) and some nucleotides [2,6,23]. Ton1914 had a high specificity of substrates, and the results are shown in Table 2. The optimum substrate for Ton1914 was sodium pyrophosphate, with a specific activity of 330 U/mg, which also showed slight phosphorolytic activity toward PPP_i_ and UTP. However, no activity was detected for ADP, ATP, or glucose-1-phosphate (Glc-1P), which implied that Ton1914 with high substrate specificity did not hydrolyze other valuable substrates (such as ATP and Glc-1P) in the one-pot UDP-sugar synthesis reaction. Although Ton1914 showed 5% relative activity toward UTP, this hydrolysis was very slight compared with that of PP_i_ and did not hinder the synthesis of UDP-sugar. We also compiled the physicochemical properties of the thermophilic PPases published thus far (Table 1), which showed that thermophilic PPases are generally excellent in terms of thermal stability as well as catalytic activity. The kinetic parameters illustrated that the K_cat_ value of Ton1914 was 2.995 × 10^4^ s^−1^ (Appendix A), which is superior to most published thermophilic PPases, whereas the affinity of the enzyme for pyrophosphate was poorest among Table 1, with a K_m_ of 1.116 × 10^−3^ M; thus, its catalytic efficiency (K_cat_/K_m_) was reduced compared to that of other PPases. In addition, the catalytic activity of Ton1914 was significantly higher than that of Ton_0002 [30] and Ton_1705 [31] from the same strain. The above results proved that Ton1914 efficiently catalyzes high-temperature pyrophosphate hydrolysis reactions.

### 2.5. Ton1914 for Improving PCR Efficiency

To systematically explore whether recombinant Ton1914 can enhance the efficiency of the polymerase chain reaction by removing PP_i_, we initially investigated the inhibitory effect of pyrophosphate on PCR. The inhibition of PP_i_ on PCR increased with increasing concentration. When the concentration of PP_i_ reached 0.2 mM, PCR was completely inhibited, while the inhibition was removed by adding 70 ng/mL Ton1914 (Figure 4a). Moreover, the length of the amplification products also affects the catalytic efficiency of PPases, and we selected six DNAs with different sizes of genes (*Tfu2902*, *Gs01315*, *Aaci0783*, *Tl08779*, *Cus2552*, and pET28a−*Tn0602*) for amplification by qPCR in the presence or absence of Ton1914. The concentration of Ton1914 in the qPCR system was determined, and the optimal level was 700 ng/mL (Appendix A). Then, the qPCR extension rate of templates after adding Ton1914 was examined. The DNA products were also assayed by agarose gel electrophoresis (AGE) after performing the same extension cycle procedure of qPCR. It is obvious that Ton1914 had a significant promotion effect on the amplification efficiency of DNA fragments of different sizes. When the same concentration of templates (5 ng/μL) was supplemented, the effect of Ton1914 on long-stranded amplification products was more favorable than that of short-stranded DNA. The yield of long-chain products increased by 32–41%, while short-chain DNA was improved by only 9.5–15% (Figure 4c and Appendix A). The results in “Appendix A” showed that although the concentration of the templates was identical, the Ct value of the amplified products gradually increased, which was due to the difference in the amplification efficiency of different genes. The amplification efficiency of long-stranded DNAs is lower than that of short-stranded DNAs, therefore more cycles are required for long-stranded amplification products to reach the fluorescence threshold. Additionally, the AGE results can prove this conclusion more visually (Figure 4b). Based on the above results, we concluded that adding Ton1914 helps to improve the amplification efficiency by hydrolyzing PP_i_.

### 2.6. Use of Ton1914 to Improve UDP-Sugar Synthesis Yield

Glucose-1-phosphate and UTP were used as substrates for the synthesis of UDP-glucose catalyzed by three UDP-glucose pyrophosphorylases (*Cs*USP, *Gk*USP, and *Tf*USP) to explore the pervasiveness of Ton1914 to help improve the yield of UDP-glucose by hydrolyzing PP_i_. The results demonstrated that Ton1914 indiscriminately promoted the catalytic efficiency of UDP-glucose pyrophosphorylases. The most significant yield improvement in synthesizing UDP-glucose was the reaction catalyzed by *Cs*USP, which presented an approximately twofold increase. The weakest was *Gk*USP, which presented only an 11% increase. This is caused by the low enzyme activity of *Gk*USP (Figure 5a). Then, *Cs*UPS was used to catalyze glucose-1-phosphate and galactose-1-phosphate (Gal-1P) to synthesize UDP-glucose and UDP-galactose and explore the effect of Ton1914 on the synthesis of various UDP-sugar. The results indicated that Ton1914 helped to increase the conversion rate of Gal-1P by only 14.5% less than that of Glc-1P, which represented an improvement of 44.05%, as Gal-1P was not the natural substrate for UDP-glucose pyrophosphorylases (Figure 5b). The above findings illustrated that Ton1914 can effectively improve the yields of UDP-glucose and UDP-galactose by hydrolyzing PP_i_ to remove the inhibition.

## 3. Discussion

Inorganic pyrophosphorylases are the main enzymes for removing PP_i_ inhibition and play an important role both for health and industrial production. For example, the drug delivery system formed by PPase-nanodiamond is expected to treat calcium pyrophosphate crystal deposition diseases and related pathologic diseases [32,33,34]. Fluorescent biosensors made by monitoring the activity of PPases can be utilized to screen for potential inhibitors of PPases, which are associated with many clinical diseases (lung cancer and colorectal cancer, etc.) [35,36,37]; In laboratory studies, using pyrophosphatase-coupled assays to monitor enzymatic activity is a potential application of PPases, such as assays of the activity of arginase, cyclic GMP-AMP synthase, and santalene synthases [38,39,40]; PPases can also be applied in industry to increase the yield of UDP-sugar synthesis and the amplification efficiency of PCR. In this work, Ton1914 exhibited excellent stability, with half-lives of 7 h and 2.5 h at 80 °C and 90 °C, respectively. The relative enzyme activity of Ton1914 incubated in Tris-HCl containing 1.6 mM Mg^2+^ at 90 °C for 5 h was still 100% of the native enzyme, while the relative enzyme activity was only 37% in the absence of Mg^2+^ after 5 h. The superb thermal stability offers outstanding potential in high-temperature pyrophosphate hydrolysis reactions. PPases have an absolute dependence on divalent metal cations, which bridge the charged reactant and enzyme nucleophile [41]. Numerous studies have shown that Mg^2+^ is an indispensable cofactor for PPases, which not only substantially increases the catalytic capacity of the enzyme but is also intimately related to the stability of PPases [6,23]. However, exceptions are observed, Ton0002, a homologous hydrolase with pyrophosphatase activity of Ton1914, was discovered by Lee [30], and it is more dependent on Ni^2+^ than on Mg^2+^. In this paper, we inferred from experiments that the appropriate amount of Mg^2+^, combined with the enzyme, enhanced the structural rigidity of the enzyme. Thus, the thermal stability and activity of the enzyme were improved.

Current research on PCR enhancers is mostly focused on inorganic molecules, such as DMSO [42], tetraalkylammonium salts [43], saccharides [16], and nanomaterials [40], while there are very few reports on PCR enhancement by PPases. To date, only Park has studied the effect of thermophilic *Ph*PPase from the archaeon *Pyrococcus horikoshii* on PCR [9], and the results showed that *Ph*PPase could increase the PCR yield with a target gene of 1.54 kbp by approximately 25%. However, his work focused on enhancing long-chain amplification products and did not systematically investigate its effect on amplifying other sizes, which requires further in-depth exploration. Approaches to improving PCR product yields using other thermophilic enzymes have been reported by Kim and Cho [11,12], who found that dITPase and dUTPase could improve PCR amplification yield by degrading dITP and dUTP. dITP and dUTP are spontaneously produced by the deamination of dATP and dCTP at high temperatures, and they stall Family B DNA polymerases by the uracil-sensing domain in the presence of deaminated bases in the template DNA. Therefore, a coupling method for the removal of pyrophosphates and degradation of deaminated nucleotides could further facilitate the PCR process and improve the yield of DNA products [9,11]. Reports have indicated that phosphoesterase domains in DNA polymerases may perform pyrophosphorylation functions and DNA polymerases containing phosphoesterase domains can couple DNA elongation and pyrophosphate hydrolysis, thus providing a mechanism for the control of the DNA extension rate and improving the efficiency of DNA elongation [17,18]. Lee first found that Ton1914 formed clusters with DNA-directed RNA polymerase subunit genes using the SSDB gene cluster search program of the KEGG [30]. This genetic structure determined that Ton1914 is a perfect coupling enhancer for DNA or RNA synthesis. The results revealed that Ton1914 had a remarkable promoting effect on PCR and was more effective for long-stranded DNA products, and they showed that it could improve the yield by up to 35–41%. This is because long-stranded DNA requires extra bases and takes longer to amplify, thus leading to the excessive accumulation of PP_i_ and more pronounced PCR inhibition. In contrast, with the addition of Ton1914, PP_i_ was degraded and the PCR amplification rate was accelerated; therefore, the enhancement of the long strand was more distinct within the same amplification cycle. 

UDP-glucose pyrophosphorylases, the key enzyme for UDP-glucose synthesis, normally catalyze the production of sugar nucleotides with poor yield. Inorganic pyrophosphatases combined with UDP-glucose pyrophosphatases can significantly improve the efficiency of sugar nucleotide synthesis by removing PP_i_ [44]. However, researchers have merely added PPases as a component to the sugar-nucleotide synthesis system, and the enhancement of PPases for the yield of UDP-sugar has never been explored separately [44,45,46]. In this paper, we investigated the ability of Ton1914 to promote the UDP-sugar synthesis reaction. The results showed that Ton1914 could boost the yield of UDP-glucose by 44.05% and UDP-galactose by 14.5%. Liu used *At*USP, *Sp*GalK and *Sc*PPase to synthesize UDP-Gal in one pot with 10 mM galactose and finally obtained 95% conversion [46]. We herein adopted the same method (*Cs*USP, *Th*GalK, and Ton1914) to synthesize UDP-Gal. The conversion rate was approximately 100% after 9 h, and the final concentration of UDP-Gal could reach 25 mM according to batch replenishment. This finding demonstrated that Ton1914 was an exceptional enhancer of sugar-nucleotide production.

Although Ton1914 has outstanding potential for industrial applications, such as DNA synthesis, gene sequencing, and sugar-nucleotide synthesis, some limitations are still observed (low substrate affinity). We predict that the efficiency of Ton1914 in hydrolyzing PP_i_ can be further improved by immobilization. Abdul and his colleagues found that graphene nanoflakes (GNFs) had unique thermal properties and enhanced PCR efficiency by increasing the thermal conductivity of the base fluids [13]. Therefore, immobilization of Ton1914 with heat-conducting nanomaterials can not only degrade PP_i_ but also increases the effective collision of substrate molecules, which will further promote PCR efficiency. Moreover, the coimmobilization of Ton1914 and *Cs*USP or the formation of Ton1914 complexes can further improve the yields of UDP-sugar. 

## 4. Materials and Methods

Materials: *Thermococcus onnurineus* NA1 (JCM13517) was purchased from JCM (Japan Collection of Microorganisms). UDP-glucose pyrophosphorylases (*Cs*USP, *Gk*USP, *Tf*USP), *Escherichia coli* BL21 (DE3) and plasmid pET28a are preserved in our laboratory. Glc-1P, Gal-1P, tetrasodium pyrophosphate decahydrate, ammonium molybdate, and Fe_2_SO_4_ were all domestic analytical purity. Acetonitrile was chromatographically pure.

Construction, expression, and purification of Ton1914: The *Ton1914* gene sequence (GenBank: CP000855.1) was retrieved from the NCBI database, and primers were designed using primer design software Primer 5.0. The PCR program settings were as follows: denaturation at 95 °C for 20 s, annealing at 65 °C for 20 s, and extension at 72 °C for 8 s. After 30 cycles, the target DNA was obtained, which was inserted into pET28a by Nco I and Xho I to obtain the recombinant plasmid. *E. coli* BL21(DE3) containing recombinant plasmid was incubated in LB liquid medium at 37 °C, and the final concentration of 0.5 mM IPTG was added when the OD_600_ reached 0.8. Incubation was then performed for 8 h at 37 °C and 120 rpm. Subsequently, *E. coli* BL21(DE3) was collected by centrifugation, resuspended in buffer (50 mM Tris-HCl, pH 7.0), sonicated, and centrifuged at 12,000 rpm for 10 min. Then, the crude enzyme was placed in a water bath at 80 °C for 10 min; the fraction of the soluble cell lysate that became denatured by heat was removed using centrifugation. Supernatant solution concentrated in ultrafiltration tubes was used to obtain the purified protein. The protein concentration was measured by the BCA protein quantification method.

Enzyme assay: In the standard assay, the enzymatic activity of Ton1914 was determined by detecting the OD_660_ of molybdophosphate heteropoly acid at room temperature [23]. The reaction system (1 mL) consisting of 2 mM MgCl_2_, 2 mM Na_4_P_2_O_7_, 50 mM Tris-HCl, and the appropriate amount of enzyme was reacted at 80 °C for 10 min and then quickly terminated on ice. The reaction mixture was blended 1:1 with the color developer to detect the maximum absorption value at 660 nm, and each group of experiments contained three parallel types. The color developer was composed of 10% ammonium molybdate solution dissolved in 5 M H_2_SO_4_ and 0.05 g/mL Fe_2_SO_4_ in 10% ammonium molybdate solution.

Unit definition of enzyme activity: the amount of enzyme required to produce 1 µmol of phosphate per minute of substrate decomposition.

Ton1914 for improving PCR efficiency: Six recombinant plasmids of different sizes, namely, *Tfu2902* (516 bp), *Gs01315* (954 bp), *Aaci0783* (1215 bp), *Tl08779* (2494 bp), *Cus2552* (3162 bp), and *pET28a*−*Tn0602* (6710 bp), were selected as DNA templates to investigate the promoting effect of Ton1914 on PCR. The amplification results were observed by real-time quantitative PCR (qPCR). The qPCR system and reaction procedure were performed according to the instructions of the PowerUp^TM^ SYBR^TM^ Green Master Mix reagent kit. In addition, each experiment was repeated three times to obtain an average value. The promotion of Ton1914 on the amplification of genes of various sizes in vitro was measured by fluorescence mapping and agarose gel electrophoresis analysis.

Ton1914s for improving the yield of UDP-sugar: UDP-glucose pyrophosphorylases (USPs) catalyze the generation of UDP-glucose (UDP-galactose) from Glc-1P (Gal-1P), and Ton1914 can accelerate the reaction forward by degrading the byproduct PP_i_. In general, all USP-catalyzed reactions were performed in a 50 mM NaH_2_PO_4_-Na_2_HPO_4_ buffer (a pH of 10.0) that contained 10 mM MgCl_2_. After 5 μg/mL USPs, 40 U Ton1914,10 mM UTP and an equimolar amount of Glc-1P (Gal-1P) had been added to the reaction buffer, the reaction was initiated at 50 °C. Each experiment included three parallel samples. The reduction in UTP was detected by HPLC at 254 nm after termination of the four-hour reaction. A 5 μm C18 column (250 nm × 4.6 nm) was used, and the mobile phase included liquid A: 12 mM tetrabutylammonium bromide, 10 mM potassium dihydrogen phosphate, and 5% acetonitrile; and liquid B: acetonitrile. Initially, A/B was reduced to 40/60 after a linear gradient from 90/10 for 8 min, and then A/B was continued at 40/60 for 20 min with a flow rate of 1.0 mL/min [47]. The yields of UDP-sugar were calculated by the consumption of UTP.

## 5. Conclusions

In this investigation, we described the thermophilic alkaline inorganic pyrophosphatase Ton1914 from *Thermococcus onnurineus* NA1 and revealed its strong substrate specificity and exceptional catalytic activity for PP_i_, which reached 389.3 U/mg. It had superior thermal and pH stability, with half-lives of 52 h and 2.5 h at 70 °C and 90 °C, respectively, and the activity remained at 80% of the maximum activity after 24 h of incubation at pH 7.0–10.0. Ton1914 is an Mg^2+^-dependent enzyme, and the relative enzyme activity of Ton1914 was still 100% after incubation in Tris-HCl containing 1.6 mM Mg^2+^ at 90 °C for 5 h. This result indicates that Mg^2+^ can improve the thermal stability of the enzyme by stabilizing the enzyme structure. Ton1914 can improve the efficiency of the polymerase chain reaction (PCR) and synthesis of UDP-sugar by hydrolyzing PP_i_ to P_i_. The results showed that Ton1914 had a significant promoting effect on PCR and improved the replication of long-stranded DNA more distinctly; moreover, when the number of bases exceeded 3000 bp, Ton1914 enhanced the yield by approximately 32%. Notably, the yield of UDP-glucose increased from 40.1% to 84.8% after the addition of Ton1914. Based on the above conclusions, Ton1914 has considerable potential for industrial applications involving DNA synthesis, gene sequencing, and nucleotide sugar synthesis.

## Figures and Tables

**Figure 1 ijms-23-12735-f001:**
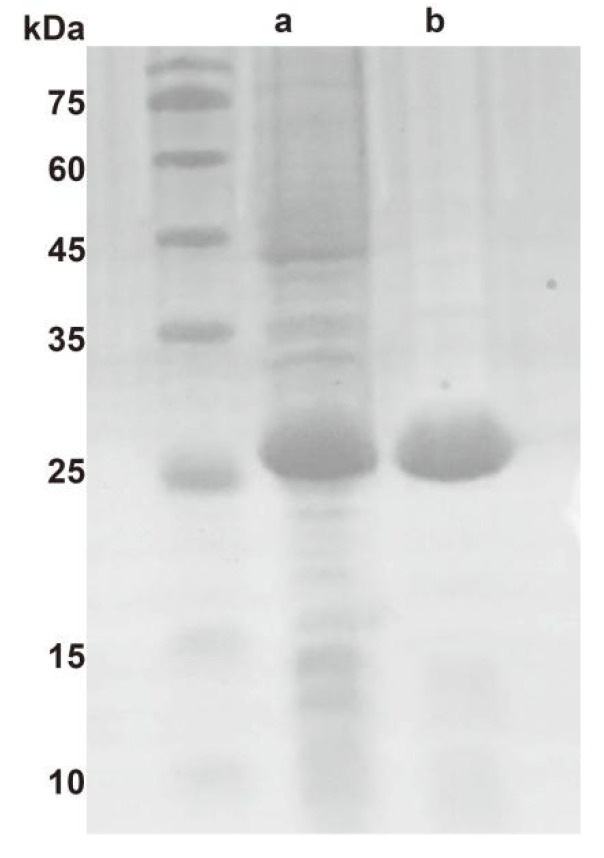
Detection of Ton1914 by SDS–PAGE. (**a**) Crude enzyme; (**b**) Purified protein heated at 80 °C for 10 min.

**Figure 2 ijms-23-12735-f002:**
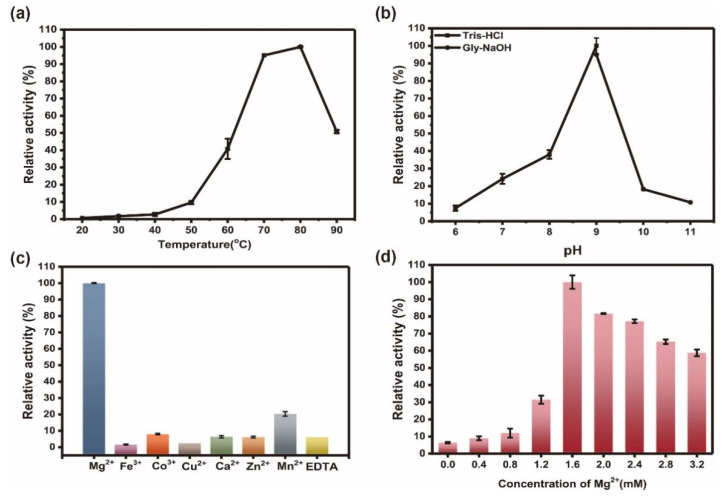
Characterization of Ton1914. (**a**) The optimum temperature of Ton1914 detected at 50 mM Tris-HCl buffer (pH 9.0); (**b**) The optimum pH of Ton1914 detected at 80 °C; (**c**) The effect of different metal ions and EDTA (final concentration 2 mM) on Ton1914; (**d**) The effect of Mg^2+^ concentration on Ton1914.

**Figure 3 ijms-23-12735-f003:**
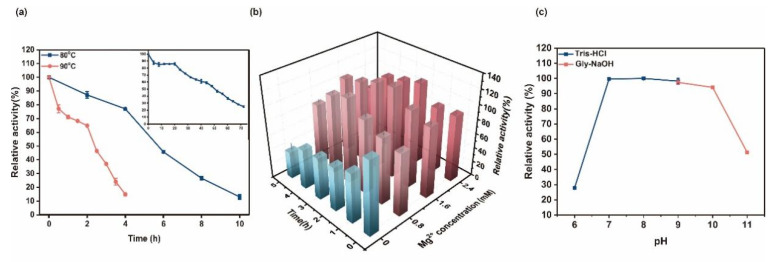
Stability of Ton1914. (**a**) The thermostability of Ton1914. The enzyme was incubated in Tris-HCl 9.0 at 70 °C, 80 °C, and 90 °C for 75 h, 10 h, and 4 h, respectively; the inset is the result of 70 °C, blue is at 80 °C, and pink is at 90 °C; (**b**) The stability of Ton1914 with Mg^2+^. The enzyme was incubated in Tris-HCl 9.0 containing 0 mM, 0.8 mM, 1.6 mM, and 2.4 mM Mg^2+^ for five hours at 90 °C. Samples were taken at hourly intervals for determining enzyme activity. The enzyme activity with 0 mM Mg^2+^ for 0 h was set as 100%; (**c**) The pH stability of Ton1914. The enzyme was incubated in Tris-HCl with different pHs at 4 °C.

**Figure 4 ijms-23-12735-f004:**
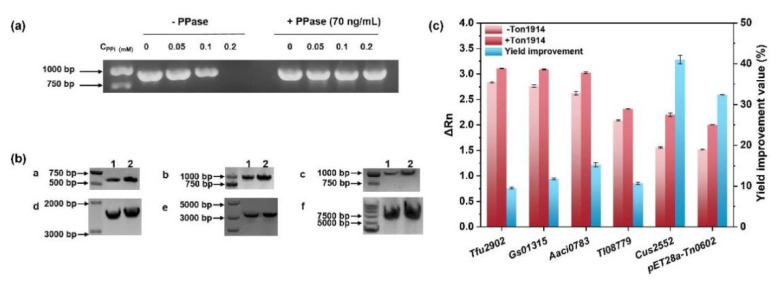
The facilitation of polymerase chain reaction (PCR) by Ton1914. (**a**) The inhibitory effect of PP_i_ on PCR, *Gs01315* (954 bp) was used as template, 0, 0.05, 0.1, 0.2 mM PP_i_ was added, and PCR was conducted with or without 70 ng/mL Ton1914, respectively. The PCR system and reaction procedure were performed according to the instructions of the PrimeSTAR^®^ HS DNA Polymerase reagent kit using conventional PCR equipment. (**b**) Agarose gel electrophoresis results of qPCR with different gene templates, a: *Tfu2902* (516 bp), b: *Gs01315* (954 bp), c: *Aaci0783* (1215 bp), d: *Tl08779* (2494 bp), e: *Cus2552* (3162 bp), f: pET28a−*Tn0602* (6710 bp); 1 was the groups without Ton1914, 2 was the groups with Ton1914; (**c**) The final fluorescence intensity of qPCR for six genes was conducted with and without Ton1914; all gene templates were amplified using the same amplification procedure and underwent 35 qPCR cycles.

**Figure 5 ijms-23-12735-f005:**
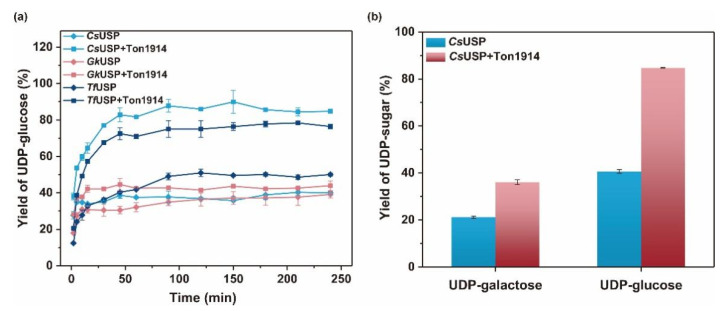
Improvement in Ton1914 for UDP-sugar synthesis. (**a**) The effect of Ton1914 on the yield of UDP-glucose catalyzed by different UDP-glucose pyrophosphorylases (*Cs*USP, *Gk*USP, *Tf*USP); (**b**) The yields of UDP-glucose and UDP-galactose catalyzed by *Cs*USP and Ton1914.

**Table 1 ijms-23-12735-t001:** Comparison of physicochemical properties of Ton1914 with other published pyrophosphatases.

Enzyme	Source of Species	pH	Temperature (°C)	t_1/2_ (h)	K_cat_ (s^−1^)	K_m_ (M)	k_cat_/K_m_ (s^−1^M^−1^)
Ton1914	*Thermococcus onnurineus* NA1	9.0	80	2.5 (90 °C)	2.995 × 10^4^	1.116 × 10^−3^	2.68 × 10^7^
PhPPase [23]	*Pyrococcus horikoshii*	7.5	70	0.83 (105 °C)	744	1.13 × 10^−4^	6.584 × 10^6^
MePPase [26]	*Methanobacterium thernoautotrophicum*	8.5	70	——	0.962 × 10^3^	1.6 × 10^−4^	6.01 × 10^6^
PfPPase [6]	*Pyrococcus furiosus*	9.5	95	46 (95 °C)	——	1.73 × 10^−4^	——
ThPPase [27]	*Thermus aquaticus*	8.3	80	——	——	6 × 10^−4^	——
AaePPase [28]	*Aquifex aeolicus*	8.0	80	1.5 (95 °C)	——	——	——
S-PPase [29]	*Sulfolobus acidocaldarius*	7.0	75	2.5 (95 °C)	1.08 × 10^3^	5.4 × 10^−6^	2.0 × 10^8^
TON_0002 [30]	*Thermococcus onnurineus* NA1	6.5	80	——	0.16	0.35 × 10^−3^	6.4 × 10^2^
TON_1705 [31]	*Thermococcus onnurineus* NA1	9.5	——	——	2.1	18.8 × 10^−6^	0.11 × 10^6^

**Table 2 ijms-23-12735-t002:** Substrate specificity of Ton1914.

Substrate	Relative Activity (%)
PP_i_	100
PPP_i_	6.9
UTP	5.0
ATP	<1
ADP	<1
Glc-1P	<1
Glc-6P	<1

## Data Availability

Not applicable.

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
