# Peer review of "Thermophilic Inorganic Pyrophosphatase Ton1914 from Thermococcus onnurineus NA1 Removes the Inhibitory Effect of Pyrophosphate"

_ijms, 2022, doi:10.3390/ijms232112735_

Round 1
Reviewer 1 Report
The authors described about the enzymatical properties of pyrophosphatase from hyperthermophilic archaeon Thermococcus onnurineus NA1 and its application for PCR and synthesis of UDP-glucose.
I have several questions and comments as follows;
1) Line 16, 37% → 37 °C
2) Line 81, Confirm ref. 21. This reference is not suitable because Shewanella is one of bacteria.
3) Line 81, archaea → archaeal
4) Line 85, PhPPase and PhoPPase are essentially same.
5) Section 2.1., Do you have any information about the native molecular mass of this enzyme and subunit structure?
6) Line 93, What pH did you use for the optimum temperature determination? What temperature did you use for the optimum pH determination?
7) Line 107, I think that the metal ions other than Mg2+ do not inhibit the enzyme activity. They just did not work as cofactor.
8) Fig. 2-4, The font size in graphs should be larger.
9) Fig. 2c, Why did you chose blue for Mg2+, Fe3+, Co2+, and Cu2+, and red for Ca2+, Zn2+, Mn2+, and EDTA?
10) Line 137, What temperature did you use?
11) Table 1, Align significant digits. Confirm the value of Km of PhoPPase.
12) Line 146-149, UTP is valuable substance (substrate) for UDP-glucose synthesis. In Table 2, the enzyme shows 5% relative activity toward UTP. The authors should mention this point.
13) Fig. 3a and c, It is difficult to read too small and blurry.
14) Fig. 3b, Please reconsider the style of the figure for better understandings.
15) Table 2, specifificity → specificity
16) Table 2, 1-P-glucose and 6-P-glucose: Are these names correct chemically?
17) Line 167, Add information about PCR enzyme (or qPCR kit) used for this experiment in this section or method section. Do you have any information about the results using different PCR enzyme or kit?
18) Line 171-179, The authors described that 70 ng/mL of Ton1914 was used in a reaction and the PCR efficiency was successfully improved (Fig. 4a). On the other hand, the authors demonstrated that the optimum concentration of Ton1914 for improving PCR efficiency was 0.7 ng/mL and excessive Ton1914 inhibited PCR (Fig. S1). How do you think about this contradiction? I think that PCR reactions are not inhibited in the presence of excess Ton1914 (Fig. S1)
19) Line 186, Figure 3c → Figure 4c?
20) Line 200, UTP should be added as substrate.
21) Fig. 5, Add the amounts of the enzymes used in the assay (UPSs and Ton1914) in the method section. How do you calculate yield of UDP-glucose? I think that determination of phosphate (Pi) in the reaction mixtures in the presence of Ton1914 helps to understand the importance of the removal of PPi on the UDP-sugar synthesis.
22) Line 272, 1914 → Ton1914
23) Line 276-278, I could not confirm the information about this sentence from ref. 44.
24) Line 286, 100%?
25) References, all scientific names should be italic style.
26) Fig. S2, Why do you think the differences in Ct values? Please show the concentration of template DNA.
Author Response
We very appreciate your suggestions for revising the article.
- Line 16, 37% → 37 °C
Response: Many thanks for the comments. This misunderstanding was caused by the linguistic statement. ‘37%’ refers to the relative activity of the control and it has been revised and marked in red in the article.
- Line 81, Confirm ref. 21. This reference is not suitable because Shewanella is one of bacteria.
Response: Thanks for the suggestion. We have reconfirmed the information of reference, which has been revised in the article.
3) Line 81, archaea → archaeal
Response: I am very sorry for this rookie mistake. The error has been corrected in the revised version.
- Line 85, PhPPase and PhoPPase are essentially same.
Response: Thanks for your suggestion. We have reconfirmed the information of PhPPase and PhoPPase and they are indeedly equal. In revised paper,We have only preserved the results of PhPPase, which was the first to be published. Additionally, we also have replaced the information of PhoPPase with PhPPase in Table 2.
5) Section 2.1., Do you have any information about the native molecular mass of this enzyme and subunit structure?
Response:Thank you for the suggestion. The theoretical molecular weight of Ton1914 was calculated in ExPASy, and its size was 20.8 kDa. Native-PAGE results showed that the molecular mass of Ton1914 is approximately 120 kDa, so it is a hexamer. We also modeled its structure using the SWISS MODEL, and the results showed that it is a homo-hexamer (The template selected is Tt-IPPase from Thermococcus thioreducens and sequence identity is 96.07 %). The revised article marked corrections to the relevant details in red.(Line 89)
6) Line 93, What pH did you use for the optimum temperature determination? What temperature did you use for the optimum pH determination?
Response: Thank you for the suggestion. In revised paper, more details are described in the captions of the Fig. 2a and Fig. 2b and highlighted in red color.
7) Line 107, I think that the metal ions other than Mg2+ do not inhibit the enzyme activity. They just did not work as cofactor.
Response: Many thanks for the comments. In revised paper, the discussion section has been described according to your advice.
8) Fig. 2-4, The font size in graphs should be larger.
Response: Thank you for the suggestion. In revised paper, all figures were modified using larger fonts.
- 2c, Why did you chose blue for Mg2+, Fe3+, Co2+, and Cu2+, and red for Ca2+, Zn2+, Mn2+, and EDTA?
Response: Thanks for your comments. To match the tone of the full-text figures, which combine pink with blue, we used different shades of pink and blue to distinguish the various metal ions in Fig. 2c. However, the picture is smaller and the colors are similar, thus it causes your confusion. In revised paper, the distinct colors of Fig. 2c were modified.
10) Line 137, What temperature did you use?
Response: Thank you for your suggestion. More details are described in the captions of the Fig. 3c and highlighted in red color. (Line 173)
11) Table 1, Align significant digits. Confirm the value of Km of PhoPPase.
Response: Very sorry for making such mistake. The error has been rectified in the revised version.
12) Line 146-149, UTP is valuable substance (substrate) for UDP-glucose synthesis. In Table 2, the enzyme shows 5% relative activity toward UTP. The authors should mention this point.
Response: Thank you for your fantastic suggestion. Although Ton1914 showed 5% relative activity towards UTP, this hydrolysis was very slight compared with that of PPi. So, it did hinder the synthesis of UDP-sugar. In revised paper, the discussion has been presented corresponding to the results. (Line 152-154)
13) Fig. 3a and c, It is difficult to read too small and blurry.
Response: Thank you for the suggestion. In revised paper, brighter colors and larger font sizes were used in Fig. 3a and c.
14) Fig. 3b, Please reconsider the style of the figure for better understandings.
Response: Thanks for your comments. This was the most appropriate form I could have come up with to express my results. I think it was the lack of details in my description of the experimental methods that caused you difficulty in understanding it. In revised paper, more details are described in the caption of the Fig. 3b with red color.
15) Table 2, specifificity → specificity
Response: I am very sorry for this mistake. The error has been corrected in the revised version.
16) Table 2, 1-P-glucose and 6-P-glucose: Are these names correct chemically?
Response: I am very sorry for this rookie mistake. The error has been corrected in the revised version with red color.
- Line 167, Add information about PCR enzyme (or qPCR kit) used for this experiment in this section or method section. Do you have any information about the results using different PCR enzyme or kit?
Response: Thanks for your comments. The information about qPCR kit was added in method section (Line 355). In addition, we also use the PrimeSTAR® HS DNA Polymerase kit to study the inhibitory effect of PPi on PCR by traditional PCR and supplemental information has been added in the caption of the Fig. 4a with red color (Line 207).
- Line 171-179, The authors described that 70 ng/mL of Ton1914 was used in a reaction and the PCR efficiency was successfully improved (Fig. 4a). On the other hand, the authors demonstrated that the optimum concentration of Ton1914 for improving PCR efficiency was7 ng/mL and excessive Ton1914 inhibited PCR (Fig. S1). How do you think about this contradiction? I think that PCR reactions are not inhibited in the presence of excess Ton1914 (Fig. S1)
Response: Thank you for your fantastic suggestion. I agree with you that the conclusion about 'excessive Ton1914 inhibited PCR' is incorrect. The contradiction you proposed is caused by the fact that 70 ng/mL and 0.7 ng/mL of Ton1914 were used under different PCR programs and PCR kits, therefore the two are not comparable. In revised paper, this conclusion was removed.
19) Line 186, Figure 3c → Figure 4c?
Response: I am very sorry for the mistake. The error has been corrected in the revised version.
- Line 200, UTP should be added as substrate.
Response: Thank you for the suggestion. In revised paper, the detail about UTP has be added with red highlight.
- 5, Add the amounts of the enzymes used in the assay (UPSs and Ton1914) in the method section. How do you calculate yield of UDP-glucose? I think that determination of phosphate (Pi) in the reaction mixtures in the presence of Ton1914 helps to understand the importance of the removal of PPi on the UDP-sugar synthesis.
Response: Thank you for the suggestion. In revised paper, the final concentration of UPSs and Ton1914 has been given in the ‘Materials and Methods’ (Line 363). The yield of UDP-sugar in this paper was calculated by consumption of UTP. Theoretically, the effect of Ton1914 is better represented by detecting phosphate production, but our experiments were performed in NaH2PO4-Na2HPO4 buffer, which could cause inaccuracy in the determination of phosphate. The yield of synthesis reaction in this buffer is the highest, that’s the reason we chose this buffer,
22) Line 272, 1914 → Ton1914
Response: We are very sorry for this mistake. The error has been corrected in the revised version.
23) Line 276-278, I could not confirm the information about this sentence from ref. 44.
Response: Thanks for the suggestion. We have reconfirmed the information of reference, which has been revised in the article with red highlight.
- Line 286, 100%?
Response: Thanks for the suggestion. ‘100%’ means that we performed a one-pot synthesis of UDP-Gal, which was catalyzed by CsUSP, ThGalK and Ton1914, using 10 mM ATP, 10 mM UTP and 10 mM galactose as substrates according to the reference 47, and the final conversion rate was 100% after 9 hours, which we misused the term "yield" to describe in the article and have corrected.
- References, all scientific names should be italic style.
Response: Thanks for the suggestion. In revised paper, all scientific names were corrected as recommended.
26) Fig. S2, Why do you think the differences in Ct values? Please show the concentration of template DNA.
Response: Thanks for the suggestion. the concentration of template DNA was showed in the revised version (Line 192). About the differences in Ct values, we added templates with the same mass concentration, therefore, the short-chain templates have more copies than the long-chain templates, and the Ct value of the short-chain DNAs are smaller compared to the long-chain product. In addition, we used qPCR to verify the promotion effect of Ton1914 on PCR, so we focused more on the variation in fluorescence intensity after adding Ton1914, and the change of Ct value was not the focus here, the final yield is.
Reviewer 2 Report
The manuscript is largely unpolished in its current form and is not suitable for evaluation. Major flaws:
1. The text is incomprehensible sometimes and is full of English editing errors.
2. References are not pertinent to the statement they are supposed to support. Sweeping or general statements are referenced by specific, narrow-focused articles.
3. Textually described findings are not supported by data (e.g. Results 2.1 should show amino acid composition or structural model to support the explanation of why the apparent molecular weight is different from the expected one)
4. Inaccurate use of basic terms, some examples: "slower electrophoretic velocity", Kcat, S-1 for secundum
5. No kinetic curves are shown to support kinetic parameters in Table 1.
6. The use of the U/mg enzyme activity term is unjustified when using purified proteins at known concentrations.
7. Low-quality figures. The axis inscriptions are small and of poor resolution, unreadable.
8. The description of the methods is insufficient and incomprehensible. E.g. "the crude enzyme was placed in a water bath at 80 C for 10 min and concentrated in ultrafiltration tubes to obtain the purified protein." I suppose the fraction of the soluble cell lysate (that the authors call crude enzyme) that became denatured by heat was removed using centrifugation, otherwise, it is not clear how this step would improve the purity of the enzyme preparation.
All in all, I could not get to the point of evaluating the scientific merit of the paper due to the rudimentary presentation.
Author Response
The manuscript is largely unpolished in its current form and is not suitable for evaluation. Major flaws:
- The text is incomprehensible sometimes and is full of English editing errors.
Response: Thanks for your comments. We have rechecked the language and editing of the entire text and corrected the editing errors in red.
- References are not pertinent to the statement they are supposed to support. Sweeping or general statements are referenced by specific, narrow-focused articles.
Response: Thanks for the suggestion. We have reconfirmed the information of all references and inappropriate references have been revised in the article with red highlight (ref. 21 and ref. 44)
- Textually described findings are not supported by data (e.g. Results 2.1 should show amino acid composition or structural model to support the explanation of why the apparent molecular weight is different from the expected one)
Response: Thank you for the suggestion. The theoretical molecular weight of Ton1914 was calculated in ExPASy, and its size was 20.8 kDa. Native-PAGE results showed that the molecular mass of Ton1914 is approximately 120 kDa. Because we characterized Ton1914 for the first time, its accurate structural information is currently not available in the PDB. While we modeled its structure using the Discovery Studio, and the results showed that it is a homo-hexamer (The template selected is Tt-IPPase from Thermococcus thioreducens and sequence identity is 96.07 %). The above results prove that the apparent molecular weight of Ton1914 is 20.8 kDa and is a homo-hexamer. The difference between the apparent molecular weight and the experimental results has been explained in our paper, which because Ton1914 carries too much positive charges (Line 86). This phenomenon has been mentioned in other articles on characterization of PPases (ref. 23). The structural information has been added to supplementary materials. (Fig. S2).
- Inaccurate use of basic terms, some examples: "slower electrophoretic velocity", Kcat, S-1 for secundum.
Response: We are very sorry for these mistakes. The error has been corrected in the revised version.
- No kinetic curves are shown to support kinetic parameters in Table 1.
Response: Thank you for the suggestion. The kinetic curves are shown in supplementary materials.
- The use of the U/mg enzyme activity term is unjustified when using purified proteins at known concentrations.
Response: Thank you for your suggestion. We approve of your suggestions and we didn’t carefully consider the accuracy of the ‘U/mg’. In revised paper, we applied the relative activity to show the change trend of enzyme activity of Ton1914.
- Low-quality figures. The axis inscriptions are small and of poor resolution, unreadable.
Response: Thank you for the suggestion. In revised paper, all figures were modified using larger fonts and higher-quality.
- The description of the methods is insufficient and incomprehensible. E.g. "the crude enzyme was placed in a water bath at 80 C for 10 min and concentrated in ultrafiltration tubes to obtain the purified protein." I suppose the fraction of the soluble cell lysate (that the authors call crude enzyme) that became denatured by heat was removed using centrifugation, otherwise, it is not clear how this step would improve the purity of the enzyme preparation.
Response: Thank you for your fantastic suggestion. We completely agree with your suggestion and the protein purification details have modified in the revised version with red. We have also added details of other experimental methods accordingly.
All in all, I could not get to the point of evaluating the scientific merit of the paper due to the rudimentary presentation.
Response: Thanks for you comments. The enzyme reported in this paper is useful, the application of patent is in progress.
Reviewer 3 Report
In my view, this is an interesting article where a new thermophilic soluble inorganic pyrophosphatase is described. While the experimental approach is quite classical and no major technical advances are described, I think that the work has been carried out rigorously. The main interest of the enzyme described is its potential biotechnological application.
The only point that must be improved is related to statistics: In all graphics experimental data points appear with error bars; however, it is not clearly stated in the manuscript the number of independent experiments performed to calculate (I guess) means and standard errors. A Student's t-test (or similar) should also be applied in Figure 5.
I would also suggest, for the sake of clarity, that the scientific names of the organisms whose PPases are shown in Table 1 appear in a legend to this figure
Author Response
In my view, this is an interesting article where a new thermophilic soluble inorganic pyrophosphatase is described. While the experimental approach is quite classical and no major technical advances are described, I think that the work has been carried out rigorously. The main interest of the enzyme described is its potential biotechnological application.
The only point that must be improved is related to statistics: In all graphics experimental data points appear with error bars; however, it is not clearly stated in the manuscript the number of independent experiments performed to calculate (I guess) means and standard errors. A Student's t-test (or similar) should also be applied in Figure 5.
Response: Thanks for the suggestion. All experiments in this paper were set up in three parallel groups, and experimental details have been added to the ‘Materials and Methods’, marked in red. Regarding the t-test, our sample size was only three and the conventional standard deviation would account for the error of the experiment. Thus, we think the average value is enough.
I would also suggest, for the sake of clarity, that the scientific names of the organisms whose PPases are shown in Table 1 appear in a legend to this figure
Response: Thanks for your comments. The source of PPases had been shown in Table 1 with red.
Round 2
Reviewer 1 Report
Dear Authors,
Thank you for your kind responses for my questions. I have an additional question.
Lines 188-191 in the revised text, you describe the optimal level of Ton1914 concentration is 0.7 ng/ml in qPCR analysis. I think that the data are derived by Fig. S3. On the other hand, it is shown in the legend that the concentrations of Ton1914 used in this assays are 0 to 0.0015 mg/ml. 0.0007 mg/ml is equal to 700 ng/ml (0.7 micro g/ml). Please tell me this point.
Line 342, Supernatatant ?
Sincerely,
Author Response
Thank you for your kind responses for my questions. I have an additional question.
Lines 188-191 in the revised text, you describe the optimal level of Ton1914 concentration is 0.7 ng/ml in qPCR analysis. I think that the data are derived by Fig. S3. On the other hand, it is shown in the legend that the concentrations of Ton1914 used in this assays are 0 to 0.0015 mg/ml. 0.0007 mg/ml is equal to 700 ng/ml (0.7 micro g/ml). Please tell me this point.
Response: We very sorry for the mistake which is caused by a lack of careful inspection. The error has been corrected in the revised version.
Line 342, Supernatatant ? Sincerely,
Response: I'm sorry for this inferior spelling mistake. The error has been corrected in the revised version, marked in red.
In addition, we rechecked the whole paper, and the following are the other errors or descriptions of inappropriate content we found, all revised in red.
Line 90 hypothesized → conjectured.
Table 1 S-1→s-1
Line 168 Deleting “which was only higher than that of Ton_1705 [31]”
Line 194 0.7 ng/mL →700 ng/mL
Line 217 Adding a description about PCR equipments
Line 236 Glc-1P→ Gal-1P
Line 287 associated with→ in
Line 310 conversion→ yield
Line 330 1-P-glucose→ Glc-1P
- P-galactose→Gal-1P
Line 385 330 U/mg→389.3 U/mg
Sincerely, thank you again for your help to improve this paper.